# Operationalization and Reporting Practices in Manuscripts Addressing Gender Differences in Biomedical Research: A Cross-Sectional Bibliographical Study

**DOI:** 10.3390/ijerph192114299

**Published:** 2022-11-01

**Authors:** Lori van den Hurk, Sarah Hiltner, Sabine Oertelt-Prigione

**Affiliations:** 1Department of Primary and Community Care, Radboud University Medical Center, 6500HB Nijmegen, The Netherlands; 2AG10 Sex- and Gender-Sensitive Medicine, Medical Faculty OWL, University of Bielefeld, 33615 Bielefeld, Germany

**Keywords:** sex, gender, report, operationalization, qualitative, methodology, reproducibility

## Abstract

Historically, authors in the biomedical field have often conflated the terms *sex* and *gender* in their research significantly limiting the reproducibility of the reported results. In the present study, we investigated current reporting practices around gender in biomedical publications that claim the identification of “gender differences”. Our systematic research identified 1117 articles for the year 2019. After random selection of 400 publications and application of inclusion criteria, 302 articles were included for analysis. Using a systematic evaluation grid, we assessed the provided methodological detail in the operationalization of gender and the provision of gender-related information throughout the manuscript. Of the 302 articles, 69 (23%) solely addressed biological sex. The remaining articles investigated gender, yet only 15 (6.5%) offered reproducible information about the operationalization of the gender dimension studied. Followingly, these manuscripts also provided more detailed gender-specific background, analyses and discussions compared to the ones not detailing the operationalization of gender. Overall, our study demonstrated persistent inadequacies in the conceptual understanding and methodological operationalization of gender in the biomedical field. Methodological rigor correlated with more nuanced and informative reporting, highlighting the need for appropriate training to increase output quality and reproducibility in the field.

## 1. Introduction

The last decade has coincided with an increasing focus on sex and gender in the biomedical field. Scientific findings are supporting the impact of sex and gender on health and disease [1], leading to a growing acceptance of the subject in the medical community. The available literature is expanding, some funding agencies are mandating the consideration of sex and gender in submitted grant proposals [2,3], and a number of scientific journals are requesting more detailed reporting of sex and gender [4,5,6].

Studying sex encompasses a focus on the impact of potential genetic, hormonal, anatomical and physiological differences on health and disease [7]. The study of gender, on the other hand, entails the study of the role of gender identity, norms, relations and institutional aspects as possible sources of health inequities [8]. Historically, sex and gender have been frequently conflated in medicine [9,10]. In most of these instances, *gender* is being used in place of *sex*, although solely biological phenomena are being studied. Critical discussions in the social sciences have highlighted the intertwined nature of sex and gender and at times rejected the concept of fixed biological sex altogether [11,12]. However, it is questionable if these complex and specific discussions have informed choices of biomedical authors. In a recent review we outlined how the operationalization of sex and gender in biomedicine is varied and lacking a concerted strategy [13]. Most instruments employed in biomedicine have been developed in the field of psychology or in the social sciences, highlighting the lack of a field-specific tradition in approaching and operationalizing the concept of gender. Sex and gender-sensitive medicine (SGSM) has emerged from a tradition of feminist health research [14], however, much of this rich theoretical basis is challenging to reconcile with the methodological standards of biomedicine as it relies heavily on quantitative positivist methods [15]. In recent years, the biomedical community is adopting an analytical approach that dissects the concept of gender into distinct dimensions, such as identity, roles or norms, relations and institutionalized gender [8]. Some instruments have been recently developed to include a combination of these dimensions [16,17,18]. Furthermore, individual quantitative investigation of the gender dimensions also appears as a promising approach with possible application in clinical practice [19]. However, no universal methodological standard has been developed to date.

Regardless of these challenges, training resources have been developed to aid researchers approaching the analysis of sex and gender in biomedicine [20]. In addition to local initiatives in teaching and training of sex- and gender-sensitive medicine [21,22], governmental agencies like the NIH Office on Womens’ Health [3], the Canadian Institutes of Health Research Institute of Gender in Health [23] and the European Commission [2] provide detailed information and online trainings. Furthermore, guidelines are available that can support researchers in reporting sex- and gender-sensitive research [24,25,26]. Regardless of the growing pool of resources, the real-world impact of these instruments on research practice remains unknown. As in many areas of healthcare, offering training resources and tools does not translate automatically into implementation in practice [27,28]. Consequently, a growing recognition of the concept of gender in biomedicine might not necessarily translate into more methodological clarity in research publications.

To address this question, we designed the present study. We aimed at investigating the operationalization and reporting practices of gender in biomedical and health research using a randomly selected sample of publications claiming the identification of gender differences. The leading questions for our mixed methods approach were:

How much methodological and operational detail do manuscripts claiming the identification of gender differences offer about the operationalization of gender? Does provision of details about the operationalization of gender correlate with more detailed reporting in other sections of the manuscript? Which examples of operationalization of gender dimensions are available in the literature?

## 2. Materials and Methods

### 2.1. Sample Selection

On 15 March 2020, we searched the biomedical database PubMed for articles claiming the identification of gender difference(s). We identified articles describing gender differences in title or abstract published in the year 2019 using the search string “gender differenc*” [Title/Abstract] AND (“2019/01/01” [ppDAT]: “2019/12/31” [ppDAT]). We exported the resulting 1568 articles into the reference manager Endnote (Clarivate, Philadelphia, PA, USA). After removal of articles that were published before or after 2019, due to, e.g., advanced online publishing, 1117 articles remained. Of these articles, a sample of 400 publications was chosen using the randomization strategy offered in the Excel program with the command “=RAND()”. The generator attributes random numbers to the whole dataset; the numbers 1–400 were selected for analysis. Ninety-eight publications were excluded (Figure 1): 40 articles were not original research (e.g., meta-analysis, review, theory, comment), for 20 no full text could be retrieved by the librarian, 16 had a spurious mention of gender, 14 were animal studies and 8 were published in a foreign language. Out of the remaining 302 articles, 69 only addressed sex differences. The remaining 233 publications full text articles were qualitatively analyzed. The articles about sex differences are excluded in the data analysis.

### 2.2. Data Collection and Analysis

Building on the SAGER guidelines [24] we developed an extended systematic search grid to allow detailed information gathering for each section of the analyzed article (Table 1). The SAGER guidelines, developed for the reporting “sex and gender equity in research”, are currently on of the two only guidelines available for the inclusion of sex and gender in biomedical research publications. The guidelines have been selected as a starting point due to their disciplinary fit and the promotion as standard by several journals [4,5,6]. For our current work, we aimed at a level of analytical detail exceeding standard reporting and have, hence, expanded the available framework. Questions were trialed and adapted in multiple rounds, before reaching the final version applied to the whole sample. The final list consists of 34 questions addressing the different sections of a scientific publication (Appendix A).

Articles were read line-by-line by two researchers (L.v.H. and S.H.) to answer the questionnaire. The two coders independently analyzed the articles in the dataset, compared answers and resolved potential disagreements by discussion with the whole team. All data were compiled in an Excel file consisting of a numerical code and text fields for qualitative answers. Answers were coded according to a detailed questionnaire (Table 1). After coding, we imported the quantitative data into SPSS version 25 (IBM Corp., Armonk, NY, USA) to compute descriptive statistics. Raw percentages are reported for the different analytical groups. Descriptive analyses are reported. We performed a comparative analysis of the manuscripts that specified the analyzed gender dimension with the ones that did not offer such detail. Gender-sensitive information was collected in each section of the manuscript (according to Table 1 and Appendix A) and then compared between the two subgroups.

For the qualitative section of the analysis, we sought information answering the following question: How can the concept of gender be operationalized in reproducible detail in biomedical publications? Again, two coders independently read the manuscript line-by-line to identify examples in the different sections of the manuscripts that reported on the operationalization of gender. Identified examples were, e.g., quotes that described theoretical background, research methods explaining the operationalization of gender or specific research instruments developed to investigate gender dimensions. Results were discussed and disagreements resolved by discussion. Data were recorded in Microsoft Excel and can be retrieved from the following OSF address: https://osf.io/k942d/ (accessed on 8 September 2022).

## 3. Results

### 3.1. Disciplinary Distribution and Authorship of the Identified Manuscripts

Of the 302 included articles, 233 addressed gender differences and 69 exclusively focused on biological sex (Figure 1). After initial characterization, we excluded the latter from further analysis. Of the articles addressing gender differences, 28.1% were published in psychology journals, 23,8% in the field of primary care and public health, 16.5% in neurology and psychiatry, 13.4% in pediatrics and internal medicine, 11.7% in social sciences and medicine, 3.9% in dentistry and surgery and 2.6% in basic science and pharmacology (Figure 2). In the group of articles addressing exclusively sex differences, 39.1% were published in the field of pediatrics and internal medicine, 20.3% in basic science and pharmacology, 17.4% in dentistry and surgery, 13.0% in neurology and psychiatry, 7.2% in primary care and public health, 1.4% in psychology and 1.4% in social sciences and management.

For 228 of the articles addressing gender differences the gender of the authors could be inferred (Figure 3). First authorship could be attributed to a man in 87 cases (38%) and to a woman in 141 cases (62%). Overall, 7 of the 228 articles were authored by a single author. Of the 221 articles with more than one author, in 106 cases (52%) the last author was assumed to be a man and in 115 cases (48%) the last author was assumed to be a woman.

### 3.2. Extent of Gender-Related Information in Manuscripts Reporting Gender Differences

We analyzed all manuscript sections (Appendix A) to identify the level of detail in reporting and potential omissions.

Overall, the analyzed abstracts mirror the detail level identified in the full manuscripts. Introduction and objectives frequently report on gender, as do results and conclusions. The methods sections oftentimes lack details about the operationalization of gender and the applied analytical approaches. Contextualization of the findings is also reported less frequently (Table 1).

In the introduction (questions I1-I5, Table 1) gender is most often addressed in the study background section and most studies (77% and 74%, respectively) report a rationale for its investigation. A specific definition of gender is rare in the analyzed manuscripts (5%), as is the reporting of preliminary data and the explicit hypothesis to be tested.

The methods sections (M1-M9, Table 1) of the manuscripts contain few methodological details. A general concept for the operationalization of gender could be found in 48% of the articles. Operationalization was performed by means of self-identification, foreign identification and document identification in line with recently reviewed approaches in biomedicine [13]. However, details about the specific gender dimensions investigated, potential power calculation to perform gender-specific analyses and the impact of gender on the research process was reported in less than 7% if the manuscripts. Gendered aspects of research ethics or access to the studies are generally not mentioned (1% and 2%, respectively).

The results sections (R1-R7, Table 1) offer most detail and gender-specific information. A total of 77% of the papers reported disaggregated numbers, 87% displayed disaggregated analyses and 71% reported disaggregated data even if no gender-related differences could be identified. A total of 89% of the manuscripts presented gender-disaggregated data in their tables and/or figures. Information about an impact of gender on study drop-outs is very limited with only 5 papers (2%) providing this information. A total of 112 (49%) of the papers report on some form of intersectional impact or analysis. The most frequently addressed categories are race/ethnicity/culture, education and socioeconomic status.

The discussion sections of the manuscripts (D1-D5, Table 1) primarily focused on offering reasons for the identified gender differences or the impact of gender overall. While 148 (64%) described implications of these findings, only 109 (47%) proceeded to reporting actionable consequences or gave remarks about the generalizability of the results. Gender as a potential limiting factor for any aspects of the study is only reported in 64 (28%) of the studies.

### 3.3. Examples of Reporting of Gender Dimensions in Biomedical Research

Of the 233 articles, only 15 provided details about the specific dimension(s) of gender being investigated. Gender identity was addressed in 6 articles, gender role in 4 articles, gender equality in 3 articles and gender stereotype, gender relations, gender norms and sex/gender in 1 article each.

The operationalization of gender in the selected publication varies, ranging from generic definitions to specific methodological operationalization. For example, Ragazan et al. [29] addressed the dimension of gender identity using a broad definition of gender.

“*We should note that we prefer the term ‘gender’ over ‘sex’ as our investigation here pertains to adults who have their identities shaped by the varying and intersecting sociocultural norms they encounter, in addition to the unique biological characteristics determined by their sex alone (Schiebinger and Stefanick, 2016)*”.(p. 184)

Perchtold et al. [30] also applied the dimension of gender identity, but described the potential interplay between sex and gender and the conscious choice to use gender as a term.

“*We adopted the current definitions of sex and gender, according to which sex is considered a biological component, which is defined via the genetic complement of chromosomes, whereas gender refers to the social, environmental, cultural, and behavioral factors and choices that influence a person’s self-identity and health (Clayton and Tannenbaum, 2016; National Institute of Health Office of Research on Women’s Health, 2019). Since it cannot be determined that any of the effects discussed in this study are caused by biological factors alone, differences between men and women are referred to as “gender differences.” This does, however, not exclude the possibility that biological and social factors may interact in explaining the present results. If cited literature addressed sex or gender differences, their wording was adopted*”.(p. 2)

A methodological definition of the dimension of gender identity was given by Luna et al. [31], who specified gender identities reported in their sample, along with the effects that gender identity may have on various aspects of health(care) and the acknowledgement of gender dimensions other than the one used in their own research.

“*Patients in our sample did not report any nonbinary gender identities. Given the health disparities, discrimination, and stigma that gender minorities experience in the health care system and the subsequent mistrust of medical professionals that arises in the community, it is possible that psychosocial-spiritual healing may be influenced by other nonbinary gender identities. Future studies should investigate the processes involved in psycho-social-spiritual healing in individuals who identify with nonbinary gender identities. In addition, we did not explore the ways in which gender roles and expectations intersect with gender identity to influence healing experiences and self-reported pain, severity of medical illness, and perception of overall health*”.(p. 1519)

Avila et al. [32] described the limitations of addressing sex and/or gender in standard cohort studies and their decision to choose “sex/gender” in the article.

“*Participants were asked whether they are male or female; however, this method does not allow us to know whether participants reported their sex or gender [31]. Thus, we will use the term “sex/gender”*”.(p. 1518).

Ahmed et al. [33], used a validated questionnaire, the Bem Sex Role Inventory [34], to operationalize the meaning of the dimension of gender that they used.

“*They included […] gender roles (using the Bem Sex Role Inventory to classify participants according to masculinity and femininity scores into the following: androgynous with both high masculinity and femininity scores; masculine with higher masculinity scores; feminine with higher femininity scores; and undifferentiated with both low scores on masculinity and femininity)*”.(p. 1200)

### 3.4. Dimensions of Gender and Representation of Gender-Related Content

We identified differences in the specificity and depth of reporting between articles that specified the investigated dimension of gender (*n* = 15) and articles that did not specify the investigated gender dimension (*n* = 218) (Figure 4). Overall, the articles providing an unambiguous and specific definition of the investigated gender dimension, offered more details and depth of analysis in other sections of the manuscript compared to the articles lacking an unambiguous definition.

At the level of the abstract, articles that specified the investigated gender dimension more frequently reported the explicit objective of conducting gender-specific research (93% and 66%, respectively) and provided significantly more gender-related information (80% vs. 34%, respectively). We could identify no differences in the methods, results and discussion sections of the abstract.

In the introduction, papers that specified the addressed gender dimension more often provided an explicit definition of gender (27% vs. 3%), reported gender-related background information (93% compared to 76%), a hypothesis on the expected gender difference (67% compared to 22%) and a substantiation of the need for gender-specific analysis (93% compared to 73%).

In the methods section, we could identify minimal differences between publications that did and did not specify the addressed gender dimension. As previously reported, the vast majority of the articles in our sample reported very few methodological details. We observed one significant difference in the reporting of the attribution of gender (self-attributed vs. attributed by other parties). In fact, 67% of articles specifying the investigated gender dimension also specified the attribution of gender compared to 46% in manuscripts not detailing the gender dimension studied.

In the results, we identified few differences between the two groups of articles. Articles reporting a specific gender dimension more often included intersectional analyses (60% compared to 48%).

In the discussion, papers that specified the investigated gender dimension reported more gender-related details overall. Differences between the groups were evident in the discussion of reasons for the identified gender differences (100% vs. 77%), implications of the gender differences (93% vs. 62%) and actionable consequences of the gender differences (80% vs. 45%).

## 4. Discussion

Our current study illustrates how manuscripts providing a clear and unambiguous definition of the investigated gender dimension offered more gender-specific information overall. Clarity about the object of investigation, thus, appears to correlate with more detailed contextualization of the results. However, the needed methodological clarity to achieve complexity and nuance is also a rare finding since only 6.5% of the analyzed publications complied with it. More strikingly, 29% of the identified publications claiming “gender differences” only analyzed and reported biological phenomena, proving how a conflation of the terminology is still a frequent phenomenon in medical research.

Overall, our study confirmed a persistent lack of specificity in the investigation of gender in biomedicine. Disciplines more aligned with concepts of the social sciences, such as public health, primary care and psychology, are more likely to effectively investigate gender, when claiming to do so. This is in line with previous research confirming more attention to the topic in these fields [13]. In the biomedical field, e.g., internal medicine, pediatrics, or pharmacology, the use of the term “gender” to describe genuinely biological mechanisms without any social implication appeared more frequent. The reasons for these differences might be multi-faceted. First, although the interest in SGSM and the knowledge about it is increasing, many researchers might still not be familiar with its analytical categories. To some authors the difference between sex and gender might indeed be unknown [9]. Second, language barriers might play a role. Many languages do not have comparable terminology to the English *sex* and *gender*, exemplifying how this differentiation could be perceived differently across cultures [35]. Hence, non-native English writers might not be aware of the theory underpinning the concepts of sex and gender in the Western academic environment [36]. Third, authors might actively choose to use “gender” instead of “sex” because they associate the inclusion of study subjects with a—more or less conscious—focus on gender identity rather than on biology [37]. Fourth, funding agencies might also adopt unspecific terminology and focus on gender representation rather than gender as an analytical content criterion without clearly distinguishing between the two [38], reinforcing the issue.

While the conscious choice to use gender instead of sex might be a possibility, the very limited number of publications providing context about their operational choices suggests otherwise. While gender identity is the most commonly addressed dimension of gender in the general media, it does not represent the whole span of what gender entails [39]. The interactional and performative aspects connected to the concept of gender [12,39], represented in the dimensions of gender norms or roles and gender relations, are not captured by solely addressing gender identity. The complexity and entanglement between different gender dimensions and, indeed, biological sex have only been explored in the biomedical field in recent years [40,41]. Although feminist critique to the medical field and its practices has a rich history [42,43], it rarely originates within medicine itself. Some authors expressed criticism at the lack of focus on gendered impacts on health in the field of SGSM, yet these are still a minority [44]. In recent years, a number of studies in the biomedical field have attempted the construction of “gender indices” by aggregating multiple items addressing the different gender domains [16,17,18]. Most of these indices were developed using binary sex as regression outcome, which might limit their robustness in identifying a “gender” construct truly distinct from biological sex. Furthermore, although promising [45], their clinical utility is still unclear. Nielsen et al. have recently followed a different approach, proposing a completely new framework for the measurement of gender as a sociocultural variable (GASV) for the health context [46]. The authors echo the established sex as a biological variable (SABV) definition employed by the National Institutes of Health (NIH) in the USA [47]. The utility of this alternative measure will also have to be established in practice. Overall, these approaches demonstrate a growing focus on the development of reproducible quantitative instruments for gender-sensitive research in the biomedical field, which should support more methodological precision in future publications.

This specificity might, indeed, improve the quality and detail of the publication. In our analysis, publications that explicitly defined the investigated gender dimension and its operationalization were more likely to provide specific background knowledge, report more details about the performed analyses, report data in a disaggregated manner even if no differences were found, and contextualize the broader implications of their findings. Several aspects might impact these findings. First, providing scientific detail about the methodological practices might correlate with a genuine focus on gender as object of investigation rather than a post hoc reporting decision [26]. Furthermore, methodological clarity might connotate specific expertise in the field of sex- and gender-sensitive analysis. Although a sex-specific analysis can be conducted by statisticians and epidemiologists [48,49,50] as detailed by several recent publications, potential limitations to the chosen approach and the analysis of the implications of the identified results might require more expert knowledge [51]. When transitioning from a sex-specific analysis to a gender-specific one, as investigated in this publication, lay understanding of the concept of gender might not suffice. Gender is a complex concept based on a rich body of knowledge mostly developed outside of the field of medicine [52]. Hence, approaching gender-sensitive analysis requires additional specific training that most physician-researchers will need to acquire [21,53,54]. Specific curricula in the field of medicine should focus on gender-related aspects in addition to sex-specific differences, ideally engaging an interdisciplinary body of lecturers. Our study clearly highlights the need for further professionalization in the field of sex- and gender-sensitive medicine with a specific focus on gender-sensitive analysis. The emergence of new instruments for analysis is promising, but their incorporation into practice will have to be supported. Funding agencies have a significant role to play in this process, as outlined by Nielsen and colleagues in a recently published framework [55]. Interestingly, the gender identity of the researcher performing the analysis appeared less relevant in our analysis than previously described [56]. Whether this could be attributed to changing authorship trends in SGSM, specific characteristics of the investigated research fields [27] will have to be established in future more extensive analyses.

To our knowledge this study is the first to provide a systematic overview of the gender analysis among publications claiming gender differences. It represents a novel contribution to the field, but several limitations need to be considered. First, our sample was a random sample and we specifically chose a relatively simple search strategy. A different random sample and more complex search strategy might have provided somewhat different results. Furthermore, we focused on a cross-sectional example of one year. We chose a year sufficiently distant from the NIH request to include sex-specific analysis [3] in research publications to see an impact on publications. However, time trends over more years might provide a different picture. Nevertheless, our findings are in line with previously described methodological inaccuracies [9] and extending our search strategy would have probably captured fewer specific publications. We assumed that publications claiming the identification of gender differences should be the ones most conscious of methodology, yet publications that failed to identify gender differences while investigating them might not have been captured by this strategy. Our search sample was limited to the English language, which might potentially represent a barrier for some authors, as previously discussed. Last, we applied the analytical concept of gender dimensions, which is currently the most common practice in medical research, however, it might not capture research using a different framework.

## 5. Conclusions

Overall, our findings highlight that many researchers still conflate the terms of sex and gender and most publications focusing on gender employ unspecific concepts and unclear methodology. The specific and unambiguous operationalization of gender, on the other hand, was associated with more detailed analysis and elaboration throughout the publication. Hence, we propose that journals requiring sex and gender reporting [4,5,6], e.g., using existing instruments such as the SAGER guidelines [24], also encourage authors to focus on methodological clarity when investigating gender. This would be in line with recent requests for more conceptual rigor in the analysis of biological sex [41,57,58]. A clear description of the object of study will validate the scientific approach and support its reproducibility, increasing the acceptance of gender-sensitive analysis in the biomedical community overall.

## Figures and Tables

**Figure 1 ijerph-19-14299-f001:**
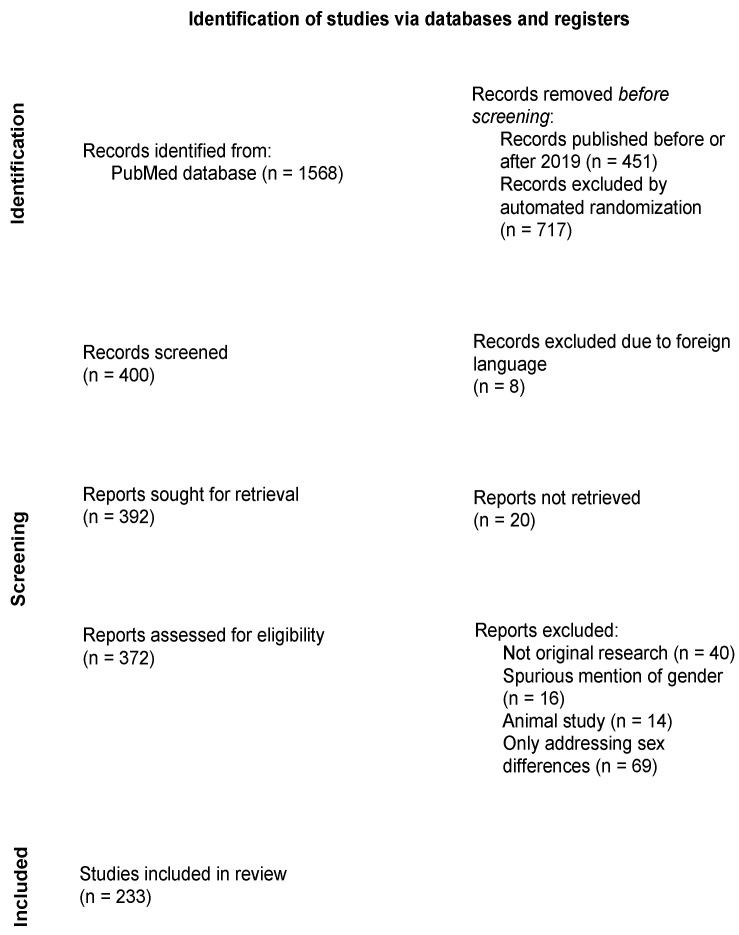
PRISMA chart of included literature.

**Figure 2 ijerph-19-14299-f002:**
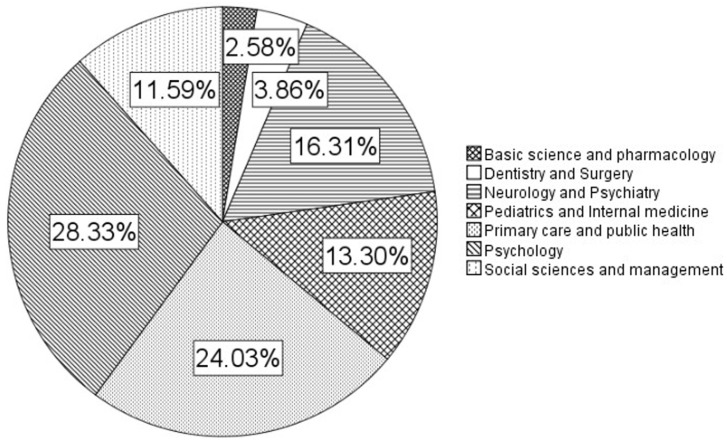
Representation of disciplinary groups and authorship patterns. All included articles were assigned to specific (bio)medical disciplines based on disciplinary affiliation of authors and MESH classifiers.

**Figure 3 ijerph-19-14299-f003:**
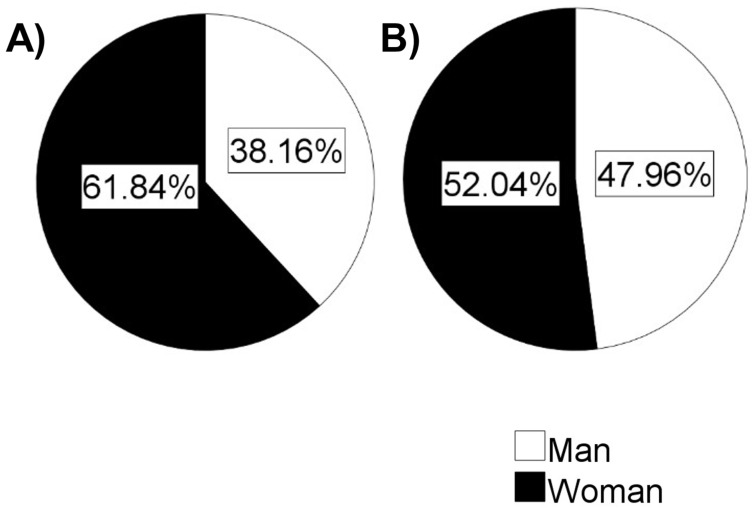
Authorship distribution. Authorship was inferred by allocation of first names, as well as web searches for identification in CV, institutional pages or social media. Authorship is reported for first (**A**) and last (**B**) authors.

**Figure 4 ijerph-19-14299-f004:**
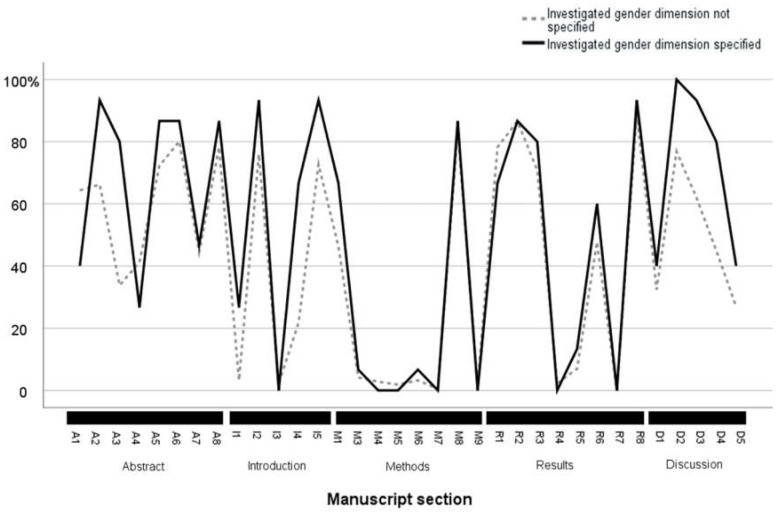
Manuscripts that specify the investigated gender dimension provide more gender-specific information overall. The figure shows the likelihood of providing gender-specific information throughout the manuscript (the specific questions for each subsection are available in Table 1 and Appendix A) in relation to the applied methodology. We compared manuscripts that specified their choices in the operationalization of gender to those that did not specify their methodological choices.

**Table 1 ijerph-19-14299-t001:** Dimensions investigated in the different sections of the manuscript.

Abstract	
A1	Structured abstract
A2	Objective: gender-sensitive research
A3	Mention of gender differences in background
A4	Details about gender identity of included participants in methods/results
A5	Gender-disaggregated reporting of results
A6	Identified gender differences (or lack thereof) addressed in the conclusion
A7	Consequences of the identified gender differences (or lack thereof) addressed
**Introduction**	
I1	Gender explicitly defined
I2	Background information about the impact of gender reported
I3	Reporting of preliminary data about gender (differences)
I4	Hypothesis-driven investigation of gender (differences)
I5	Need for gender-sensitive analysis substantiated
**Methods**	
M1	Attribution of gender explicitly described
M2	Gender dimension(s) addressed
M3	Consideration of gender upon recruitment
M4	Statistical power to analyze the impact of gender reported
M5	Impact of gender on study access described
M6	Impact of gender on the functioning of the study/intervention reported
M7	Gender distribution in the research team mentioned
M8	Gender-disaggregated analysis presented
M9	Consideration of gender-specific ethical aspects
**Results**	
R1	Included numbers reported by gender (identity)
R2	Analyzed data disaggregated by gender (identity)
R3	Gender-disaggregated reporting even if no differences found
R4	Drop-outs, withdrawals, outliers, loss-to-follow up reported by gender (identity)
R5	Gender-specific confounders mentioned and corrected for
R6	Intersectional analyses included
R7	Gender differences represented in tables, figures and/or graphs
**Discussion**	
D1	Considerations towards the generalizability of gender-sensitive results included
D2	Reasons for gender differences included
D3	Implications of gender differences discussed
D4	Actionable consequences of gender differences discussed
D5	Gender-specific limitations reported

## Data Availability

Raw data and codebook can be accessed at the following address: https://osf.io/k942d/ (accessed on 8 September 2022).

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
