# Peer review of "Operationalization and Reporting Practices in Manuscripts Addressing Gender Differences in Biomedical Research: A Cross-Sectional Bibliographical Study"

_ijerph, 2022, doi:10.3390/ijerph192114299_

Round 1
Reviewer 1 Report
This article presents a study where, through one systematic research, the authors “investigated current practices of reporting on gender in biomedical publications that claim to identify “gender differences””.
Overall, the article seems to me to be very well written and clear, and I only have four minor revisions to point out:
In section 3. “Results”, the results presented in the first and second paragraphs, between lines 106 and 119, are not the same as the results presented in Figure 2, and Figures 2.2a and 2.2b. Although the numbers are rounded in the text, differences are noticeable, and this is very confusing. I suggest that a careful review of these results is made.
I think Figures 2.2a and 2.2b would look better and clearer if they were separated from Figure 2 (e.g., being called Figure 3).
It is not clear what kind of analysis was done in Figure 3 exactly. Please clarify.
Finally, section 4. "Discussion" needs to be improved in order to stop being just a mere synthesis of the results and become a real discussion of the results in the light of theory. For example, page 10 of the text currently has only one reference, which is very little.
Reviewer 2 Report
This research paper sought to explore the current reporting practices around gender in biomedical publications that claim the identification of “gender differences”. Drawing on a “qualitative” analysis of 302 articles the paper assesses the provided methodological detail in the operationalization of gender and the provision of gender-related information throughout the manuscripts. Findings show that whereas just 23% solely addressed biological sex, only 15 articles of those investigating gender offered reproducible information about the operationalization of the gender dimension studied. The study demonstrates that many researchers still conflate the terms of sex and gender and most publications focusing on gender employ unspecific concepts and unclear methodology. As they observed persistent inadequacies in the conceptual understanding and methodological operationalization of gender, the authors highlight the need for appropriate training to increase output quality and reproducibility in the biomedical field.
I believe the proposed paper is timely as: (a) it focuses on a central issue - gender in medical science – for growing equality and diversity in research; (b) it draws on an significant review of research literature, showing patterns of how gender is addressed by physician-researchers; (c) it goes beyond a broad descriptive analysis of results by either exploring the extent of gender-related information in the different sections of the manuscript or presenting examples of reporting of gender dimensions in the field.; (d) and, as authors in the biomedical field have significantly limited the reproducibility of the reported results by often conflating the terms sex and gender in their research, it contributes to sensitize researchers, journals with peer review, funding agencies and other stakeholders by highlighting the need to further gender knowledge in the field. Nonetheless, the proposed paper is presented in a report format, hence unsuitable for publishing in a scientific journal. For the reasons outlined below, I think that the manuscript requires a thorough revision in order to both make the transition from this format to one more adjusted to a scientific article, and provide it to draw out some original contribution to gender research in the biomedical field. While the authors claim that “this study is the first to provide a systematic overview of the gender analysis among publications claiming gender differences”, I do think that without a major revision the paper will not make the intended contribution to literature on how to make medical research gender-sensitive.
1.
First and foremost, the authors must introduce a section that goes further in contextualizing the object of investigation, that is how gender has been incorporated in biomedical research. In this section, they should address central issues such as why medical literature historically have conflated the terms of sex and gender, or why most publications focused on gender employ unspecific concepts and unclear methodology despite the several training resources that have been developed to aid researchers approaching the analysis of sex and gender in biomedicine. Hence, the paper requires a deeper state of the art regarding gender in medical science to raise the main research questions and hypothesis so as to make a more substantial discussion as far as findings are concerned.
2.
The authors must also present a theoretical section that both overviews the literature and the main strands stemming from gender studies and feminist theories on gender, not only to go theoretically further in discussing gender in science, but to justify more accurately why the use the concept of “gender dimensions” and better clarify upon which theoretical foundations it draws. In doing so, the authors would: (a) present the reasons for both introducing the gender lens in the biomedical research and discussing the concept in its great complexity (as they properly say, “Gender is a complex concept based on a rich body of knowledge mostly developed outside of the field of medicine”); (b) discuss the several consequences of conflating sex and gender in medicine; and (c) demonstrate the virtues of going beyond a lay understanding of the concept of gender by exposing their own background knowledge on the subject. In sum, they must present a research framework that allows them to go beyond the observation of the operationalization of gender dimensions in the literature, the methodological and operational detail, and correlation between the provision of details about the operationalization of gender and more detailed reporting in other sections of the manuscript.
3.
There are several issues to be addressed in both Methods and Results sections. First, why did the authors only search for articles published in 2019? As the paper is exhaustive in presenting percentages in the Results section, I wonder if: (a) a sample of 302 articles is of sufficient size to go further in the descriptive statistics; (b) addressing a much larger time range (e.g. from 2010 to 2020) would allow to make more consistent findings regarding current trends; Second, does the study intend to provide data that would be representative of trends in research literature in the biomedical field? If yes, this is possibly true only for 2019. Otherwise, is this just a preliminary study whose aim is to pave the way for a more comprehensive research that would provide an accurate complete overview on the trends overtime? Or is the aim of this exploratory study to just go deeper in the content analysis of the gathered publications? If so, why did the authors systematically address in a descriptive quantitative fashion the data randomly selected for detailed qualitative analysis? Why did they not choose to a truly qualitative strategy enabling an analysis that goes beyond the operationalization of gender and the provision of gender-related information throughout the manuscript, addressing: how different the approaches towards gender are; what type of research questions physician-researchers have raised as gender is concerned; and, as for raised gender questions and approaches, what qualitative differences in biomedical literature claiming the identification of gender differences the authors observed? In sum, the authors must properly clarify the applied methodological strategy and the reasons for choosing it as well to substantially improve the Results section, which prioritises a quantitative approach and presents some pointless detailed findings, such as “Of the 279 articles with more than one author, in 148 cases (53.0%) the last author was assumed to be a man and in 131 cases (47.0%) the last author was assumed to be a woman” (p. 5). For an example of a recent literature review applying a critical qualitative analysis of the literature in the academic context, see: Rodrigo Rosa (2022) The trouble with ‘work–life balance’ in neoliberal academia: a systematic and critical review, Journal of Gender Studies, 31:1, 55-73, DOI: 10.1080/09589236.2021.1933926
4.
I do think that, as above suggested, the introduction of a contextual section and an informed theoretical discussion about gender in science would lead to a more fruitful debate in the Discussion and Conclusion sections. First, I think the opening observation “Our current study illustrates how manuscripts providing a clear and unambiguous definition of the investigated gender dimension offered more gender-specific information overall.” (p. 9) is a rather tautological finding, as in any research the richness of the gathered information is quite dependent on the clarity invested in the definition of the object of study and, thus, in the dimensions of the concept to be investigated. But more importantly, the presented discussion does not go beyond the observed relationships between qualitative variables through a quantitative analysis. What challenges do the authors point out while observing the relentless propensity of physician-researchers’ to conflate the terms sex and gender in their research? This type of questions must be raised taking into account the socio-historical background, and particularly the fact that funding agencies are increasingly mandating the consideration of sex and gender in submitted grant proposals. What do the findings suggest as far the role of the State, and specifically organizations fighting for equality in science, in tackling gender inequality in the biomedical field? The authors claim that the “study clearly highlights the need for further professionalization in the field of sex- and gender-sensitive medicine and recognition of the specific expertise related to the performance and contextualization of these analyses”. However, in the brief Conclusion section they neither present nor elaborate recommendations to increase output quality and reproducibility in the field. Like their discussion remarks, their Conclusions lack ambition. To sum up, in what sense would this discussion contribute to improving the translation of training resources and tools into implementation in practice?
5.
In the Abstract: “Our systematic research identified 1117 articles for the year 2019. Four hundred of these were randomly selected for detailed qualitative analysis and 302 matched inclusion criteria.” As they present several descriptive statistics, I think the authors should be more straightforward in mentioning database length, which is of 302 publications.
6.
The following sentence lacks references: “Critical discussions in the social sciences have highlighted the intertwined nature of sex and gender and at times rejected the concept of biological sex altogether.” (p. 1).
7.
For the sake of clarity, elaborate what is the “randomization strategy offered in the Excel program” (p. 2).
8.
Clarify in the body text what are the SAGER guidelines, and elaborate on the reasons for choosing them; to simply add the respective references is not enough.
9.
Please clarify the acronym SGSM (p. 9).
Round 2
Reviewer 2 Report
The paper was very much improved with all the changes the authors have introduced according the reviewers comments. It is clear for me that the authors make enough effort to turn the paper into a publishable version. Hence, I think it should be accept for publication.